

# The Lituya Bay landslide-generated mega-tsunami. Numerical simulation and sensitivity analysis

José Manuel González-Vida[1], Jorge Macías[2], Manuel Jesús Castro[2], Carlos Sánchez-Linares[2], Marc de la Asunción[2], Sergio Ortega-Acosta[3], and Diego Arcas[4]

[1]Dpto. de Matemática Aplicada, ETSII, Universidad de Málaga, 29080, Málaga. Spain
[2]Dpto. de A.M., E. e I. O. y Matemática Aplicada, Facultad de Ciencias, Universidad de Málaga, 29080, Málaga. Spain
[3]Unit of Numerical Methods, SCAI, Universidad de Málaga, 29080, Málaga. Spain
[4]NOAA/Pacific Marine Environmental Laboratory (PMEL), Seattle, WA, USA

*Correspondence to:* J. Macías jmacias@uma.es

**Abstract.** The 1958 Lituya Bay landslide-generated mega-tsunami is simulated using the Landslide-HySEA model, a recently developed finite volume Savage-Hutter Shallow Water coupled numerical model. Two factors are crucial if the main objective of the numerical simulation is to reproduce the maximal run-up, with an accurate simulation of the inundated area and a precise re-creation of the known trimline of the 1958 mega-tsunami of Lituya Bay. First, the accurate reconstruction of the
initial slide. Then, the choice of a suitable coupled landslide-fluid model able to reproduce how the energy released by the landslide is transmitted to the water and then propagated. Given the numerical model, the choice of parameters appears to be a point of major importance, this leads us to perform a sensitivity analysis. Based on public domain topo-bathymetric data, and on information extracted from the work of Miller (1960), an approximation of Gilbert Inlet topo-bathymetry was set up and used for the numerical simulation of the mega-event. Once optimal model parameters were set, comparisons with observational
data were performed in order to validate the numerical results. In the present work, we demonstrate that a shallow water type of model is able to accurately reproduce such an extreme event as the Lituya Bay mega-tsunami. The resulting numerical simulation is one of the first successful attempts (if not the first) at numerically reproducing in detail the main features of this event in a realistic 3D basin geometry, where no smoothing or other stabilizing factors in the bathymetric data are applied.

## 1  Introduction

Tsunamis are most often generated by bottom displacements due to earthquakes. However, landslides, either submarine or subaerial, can also trigger devastating tsunami waves. Besides they are, on some occasions, extremely destructive as they form near the coast or in the same coastline if they are aerial. Sometimes landslides may generate so-called mega-tsunamis which are characterized by localized extreme run-up heights (Lituya Bay, 1958 Miller (1960); Fritz et al. (2009); 1934 Tafjord event, Norway Jørstad (1968); Harbitz et al. (1993), Taan Fjord, October 17, 2015 Bloom et al. (2016); Higman et al. (2017), among many
others). For seismic tsunami simulations, in general, the most critical phases are generation and arrival to a coast, including inundation. Propagation over deep basins can be modeled using the non-linear shallow water (NLSW) equations or more typically using a non-diffusive linear approximation. With landslide generated tsunamis, however, matters get more complicated.





The generation phase itself becomes critical and complex effects between the landslide and the water body must be taken into account. Most notably, the case of subaerial landslide generated tsunamis is where modeling and numerical implementation becomes most critical, owing to these events producing more complex flow configurations, larger vertical velocities and accelerations, cavitation phenomena, dissipation, dispersion and complex coupled interaction between landslide and water flow.

It is evident that shallow water models cannot take into account and reproduce all of these phenomena, particularly vertical velocities or cavitation. We do, however, demonstrate here that such models can, despite limitations, be useful for hazard assessment in which the main features (from a hazard assessment point of view) of these complex events, such as runup and main leading wave, are reproduced. The overall aim is not to accurately reproduce the evolution of the displaced solid material or the dispersive nature of the trailing waves as the perturbation propagates, but, alternatively to accurately reproduce the impact

of tsunami waves to coasts in terms of runup and flood area. Comparison of the numerical results with the observed trimline presented here are shown to support our statement that: a fully coupled, vertically integrated Shallow Water/Savage-Hutter model can, effectively and accurately, reproduce the runup and coastal inundation resulting from aerial landslides generated in fjords and enclosed basins. This study further supports this assessment by comparing model results with observed data for a paradigmatic example of extreme runup produced by an aerial landslide in an enclosed bay, a simulation that has not been

successfully undertaken previously with more comprehensive numerical models.

Full 3D numerical modeling of landslide generated tsunamis Horrillo et al. (2013) is uncommonly used for real world scenarios due to the highly demanding computational resources required. Whereas common thought persists that NLSW equations are sufficient for simulation of ocean-wide tsunami propagation averaged models, the importance of frequency dispersion for modeling landslide generated tsunamis lends preferrence to Boussinesq models. In no case, however, can any of these standard

models describe the violent impact of subaerial landslides with flow separation and complex subsequent flow patterns and slide material evolution. We further claim that, in enclosed basins, the two main mechanisms that need to be well captured and accurately reproduced by a numerical model are, first the transmission of energy from the slide material into the water body and, second, the coastal inundation by means of accurate wet/dry treatment. Confinement makes the role of dispersion minor relative to other effects.

In the case of tsunamigenic aerial landslides in fjords, bays, or any long and narrow water body, confinement and reflection (a process that also makes propagation and interaction more complex) are relatively more important considerations than dispersion which becomes less important. This is particularly true for the leading wave Løvholt et al. (2015) that, on the other hand, is mainly responsible for coastal impact. Lindstrøm et al. (2014) using a scaled laboratory set up showed that wave propagation along the fjord involved frequency dispersion but only to a moderate extent Løvholt et al. (2015). It is in the far field where

dispersive effects are proven to be important for a realistic description of tsunami impact Løvholt et al. (2008); Montagna et al. (2011).

Despite this, and independent of the eventual confinement of the flow, many authors continue to claim the absolute need for dispersive, or even full Navier-Stokes models Abadie et al. (2009), when dealing with the simulation of landslide generated tsunamis. Still other authors, not so strict in their assessment, claim that dispersive models represent a better alternative than

NLSW models Løvholt et al. (2015). The lack of dispersive model simulations in the literature of the Lituya Bay event play



against the argument of ruling out non-dispersive coupled models, such as the one proposed in the present work. Very recently, in February 2017 during the "2017 U.S. National Tsunami Hazard Mitigation Program (NTHMP) Tsunamigenic Landslide Model Benchmarking Workshop", held on the Texas A&M Galveston, Texas campus participants agreed to recommend to the NTHMP Mapping and Modeling Subcommitee (MMS) the use of dispersive models for numerical simulation of landslide

generated tsunamis. For the case of enclosed basins, bays, and fjords, it was agreed that NLSW models remain a suitable tool.

Among all examples of subaerial landslide generated tsunamis, the Lituya Bay 1958 event occupies a paradigmatic place in the records, standing alone as the largest tsunami ever recorded and representing a scientific challenge of accurate numerical simulation. Based on generalized Froude similarity, Fritz et al. (2009) built a 2D physical model of the Gilbert inlet scaled at 1:675. A number of works have focused their efforts in trying to numerically reproduce Fritz et al. (2009) experiments Mader

and Gittings (2002); Quecedo et al. (2004); Schwaiger and Higman (2007); Weiss et al. (2009); Basu et al. (2010); Sánchez-Linares (2011) Detailed numerical simulations of the real event in the whole of Lituya Bay with a precise reconstruction of the bottom bathymetry and surrounding topography are limited. As an example, Mader (1999) fails in reproducing the 524 meter run-up and concludes that the amount of water displaced by a simple landslide at the head of the bay is insufficient to cause the observed tsunami wave. As far as we know, the present work represents the first successful attempt to realistically simulate and

reproduce the 1958 Lituya Bay mega-tsunami in a realistic three dimensional geometry with no smoothing in the geometry or initial conditions. The topobathymetric data are the raw data available interpolated in the computational mesh, with no further treatment, and the initial impact is not smoothed and no additional numerical treatment is required.

The aim of this work is to produce a realistic and detailed simulation of the Lituya Bay 1958 mega-tsunami. The Landslide-HySEA model Castro et al. (2005, 2006); Gallardo et al. (2007); Castro et al. (2008, 2012); Macías et al. (2012); de la Asunción

et al. (2013); Macías et al. (2015) developed by the EDANYA research group (https://edanya.uma.es) is used for this simulation. HySEA models have been fully validated for tsunami modeling using all of the benchmark problems posted on the NOAA NTHMP web site for propagation and inundation Macías et al. (2017b), for tsunami currents Macías et al. (2018a, 2017d) and for landslides Macías et al. (2017a). In Sánchez-Linares (2011), the Landslide-HySEA model is used to reproduce the Fritz et al. (2009) laboratory experiment, producing good results. In the present work, a detailed full-scale three-dimensional

benchmark experiment of Lituya Bay studying the tsunami generation, propagation and runup in several relevant areas is performed. The numerical results presented here demonstrate that a Savage-Hutter model for the slide material fully coupled with Shallow Water equations for the water flow can suitably reproduce the main features of an extreme event such as the 1958 Lituya Bay landslide generated tsunami.

## 2 Background

At 06:16 UTC on July 10, 1958, a magnitude $M_w$ 8.3 earthquake occurred along the Fairweather Fault (Alaska, USA). This quake triggered a landslide of approximately 30.6 km$^3$ in Gilbert Inlet Miller (1960) that in turn produced the largest tsunami run-up ever recorded Fritz et al. (2009). The epicenter of this quake was a scant 21 km from Lituya Bay. Intense shaking lasted for 1 to 4 minutes according to two eyewitnesses who were anchored at the entrance of the bay. According to Miller (1960),

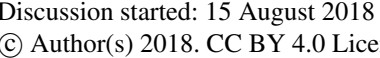
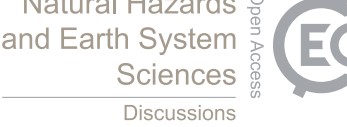


between 1 and 2.5 minutes after the earthquake, a large mass of rock slid from the northeast wall of Gilbert Inlet. It is probable that this entire mass of rocks, ice, and soil plunged into Gilbert Inlet as a unit. The result was the sudden displacement of a large volume of water as the slide was plunged into Gilbert Inlet causing the largest tsunami ever evidenced. The upper limit of destruction by water of forest and vegetation (known as trimline) extended to a maximum of 524 m above mean sea level on

the spur southwest of Gilbert Inlet (Figure 1). Maximum inundation distance reached to 1,400 m on flat ground at Fish Lake on the north side of the bay, closer to its entrance.

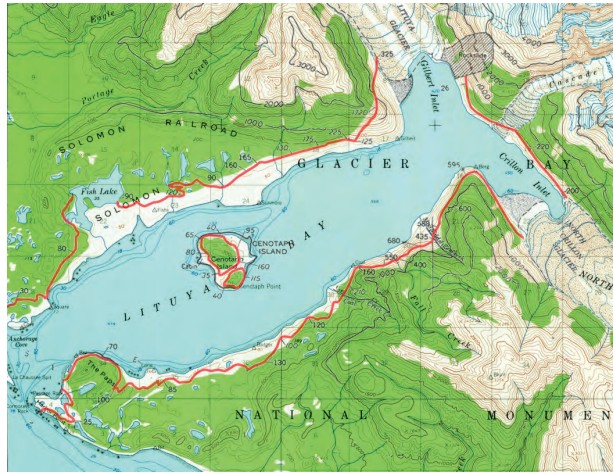

**Figure 1.** Topographic map of Lituya Bay (U.S. Geologic Survey, 1961) showing the settings and trimline of 1958 megatsunami (based on data from Miller (1960)). Units in feet. Key locations as Gilbert and Crillon Inlets, Cenotaph Island, Fish Lake or The Paps are shown.

In order to understand the evolution of the giant Lituya Bay wave, a rough model at a 1:1,000 scale was constructed at the University of California, Berkeley (R.L. Wiegel in Miller (1960)). If the slide occurred rapidly as a unit, they concluded that a sheet of water washed up the slope opposite the landslide to an elevation of at least three times the water depth. At the same

time, a large wave, several hundred feet high, moved in a southward direction, causing a peak rise to occur in the vicinity of Mudslide Creek, on the south shore of Gilbert Inlet. According to Miller (1960), this peak reached 204 m (580 ft) (see Figure 1).

The landslide was triggered by fault movement and intense earthquake vibrations Fritz et al. (2009) and it is highly probable that the entire mass of rock plunged into Gilbert Inlet as a unit, as previously stated. Nevertheless, there is no consensus about

the typology of the slide mass movement. Miller (1960) provides discussion setting this event near the borderline between a landslide and a rockfall following the classifications of Sharpe (1938) and Varnes (1958) while Pararas-Carayannis (1999) classified the mass movement as subaerial rockfall, making the distinction from gradual processes of ordinary landslides. Nevertheless, as will be shown in the next sections, in Fritz et al. (2001) and Fritz et al. (2009), the authors proposed and, in fact show, a landslide typology that, based on the generalized Froude similarity, make it possible to reproduce this event using a

two-dimensional scaled physical model of the Gilbert Inlet. A pneumatic landslide generator was used to generate a high-speed



granular slide with density and volume based on Miller (1960) impacting the water surface at a mean velocity of 110 m/s. The experimental runup matches the trimline of forest destruction on the spur ridge in Gilbert Inlet.

## 3 Area of Study

Lituya Bay (Figure 1), located within Glacier Bay National Park on the northeast shore of the Gulf of Alaska, is a T-shaped tidal inlet, nearly 12 km long and with width ranging from 1.2 to 3.3 km except at the entrance, where the width is approximately 300 m. The north-eastward stem of the bay cuts the coastal lowlands and the foothills flanking the Fairweather Range of the St. Elias Mountains. In the vicinity of the head of the bay, the walls are steep fjord-like and rise to elevations ranging from 670 m to 1,040 m in the foothills inmediately to the north and south, and more than 1,800 m in the Fairweather Range. In 1958 the maximum depth of the bay was 220 m and the sill depth, at the entrance of the bay, was only 10 m. At the head of the bay, the two arms of the T, the Gilbert (northern arm) and Crillon (southern arm) Inlets, form part of a great trench that extends tens of kilometers to the northwest and southeast on the Fairweather fault. Cenotaph Island divides the central part of the bay into two channels of 640 m and 1,290 m, respectively.

### 3.1 Coastal morphology

The shores around the main part of the bay are composed mainly of rocky beaches that rise steeply away from the shoreline. There are two adjoining land masses that rise away from the beach, ranging in elevations from less than 30 m at a horizontal distance of 2 km, around Fish Lake, to 170 m at a horizontal distance of 370 m at The Paps (see Figure 1). Prior to the 1958 tsunami, low deltas of gravel had built out into Gilbert Inlet along the southwest and northeast margins of the Lituya Glacier front.

According to Miller (1960), and as evidenced in several graphical documents, after the tsunami, the delta on the northeast side of Gilbert Inlet completely disappeared, and that on the southwest side of the bay was noted to be significantly smaller. To re-create the scenario previous to the event for use in the numerical simulation presented here, shorelines, deltas, and the glacial front inside Gilbert Inlet before 1958 were all taken from Miller's reconstruction (see Figure 2).

## 4 Bathymetric Data

According to Miller (1960), examination of Lituya Bay bathymetry was the first step in determining whether the volume of water was sufficient to account for a 524 m wave. Bathymetric surveys made in 1926 and 1940 U.S. Coast and Geodesic Survey (1926), show that the head of Lituya Bay is a pronounced U-shaped trench with steep walls and a broad, flat floor that slopes gently downward from the head of the bay to a maximum depth of 220 meters just south of Cenotaph Island. At this maximum depth, the slope then rises toward the outer part of the Bay. At the entrance of the Bay, the minimum depth is on order 10 m during mean lower low water. The outer portion of Lituya Bay is enclosed by a long spit, the La Chaussee Spit, with only a very narrow entrance of about 220-245 m that is kept open by tidal currents. The tide in the bay is predominantly diurnal, with



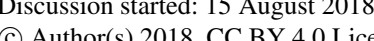


**Figure 2.** Gilbert Inlet shoreline and glacier, bathymetries and shorelines before and after 1958. (A) Modified location of shoreline and glacier front before 1958, from Miller (1960); (B) Shoreline and glacier front of Lituya Glacier after the 1958 tsunami; (C) Reconstructed Gilbert Inlet bathymetry (based on 1926 data); (D) Gilbert inlet bathymetry (1959).

a mean range of 2 m and a maximum range of about 4.5 m U.S. Coast and Geodesic Survey (1959). The U-shape of the bay coupled with the flatness of its floor indicate that extensive sedimentation has taken place. The thickness of the sediments at the Gilbert and Crillon inlets is not known, but believed to be substantial due to terminal moraine deposition during different brief glacial and interglacial episodes.

5    The bathymetric data used for the modeling work presented here were obtained from the U.S. National Ocean Service: Hydrographic Surveys with Digital Sounding. Data from Survey ID: H08492, 1959, were used as reference bathymetry since





this survey is the nearest in time to the data of the tsunami and there were enough data collected to provide a good representation of the entire Lituya Bay seafloor. Data from Survey ID: H04608,1926 were used to reconstruct Gilbert Inlet bathymetry as these were the closest pre-event data available. Unfortunately, data from this survey are not sufficient in resolution to provide an acceptable bathymetric grid for our study of the entire bay. Nevertheless, the survey provides both enough data and detailed

information of pre-tsunami bathymetry in the Gilbert Inlet. In Figures 2.c and 2.d it is shown the 1926 reconstructed bathymetry and the original 1959 bathymetry, respectively. The mass volume difference between these two bathymetries in Gilbert Inlet area is about $31 \times 10^6 \, \mathrm{m}^3$, that is, very close to the slide volume estimated by Miller (1960).

## 5   Tsunami source

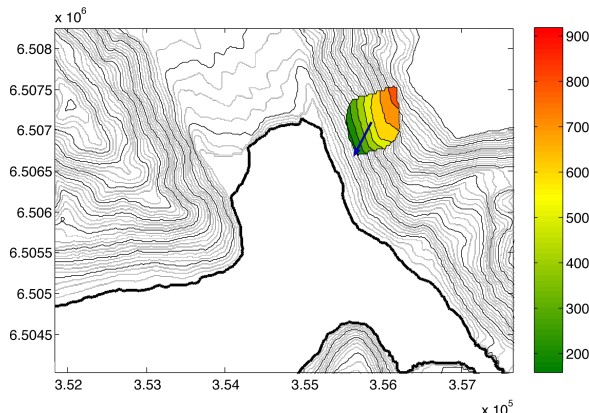

**Figure 3.** Initial conditions for the slide: Slide location and initial velocity vector direction.

The dimensions of the landslide on the northeast wall of Gilbert Inlet were determined with reasonable accuracy by Miller

(1960), but the thickness of the slide mass normal to the slope could be estimated only roughly from available data and photographs. The main mass of the slide was a prism of rock that was roughly triangular in cross section, with dimensions from 730 m to 915 m along the slope, a maximum thickness of about 92 m normal to the slope and a center of gravity at about 610 m elevation. From these dimensions the volume of the slide estimated by Miller (1960) was of $30.6 \times 10^6 \, \mathrm{m}^3$.

To locate and reconstruct the volume of the slide mass the following procedure was implemented: First, based on aerial

photos and data provided by Miller (1960), the perimeter of the slide was determined. Then, an approximate centroid for the formerly defined surface was considered drawing two lines, one horizontal and another one vertically projected on the surface. The surface centroid was located at 610 m high, defining the upper bound for the mass slide. The volume of the reconstructed slide was of $30.625 \times 10^6 \, \mathrm{m}^3$, which matches accurately with Miller's estimation. The three criteria we tried to fulfill in order to reconstruct the slide geometry were: (1) to place it in its exact location (projected area); (2) keep an approximate location

for the centroid, also in height; and (3) to recover an accurate volume for the numerical slide.





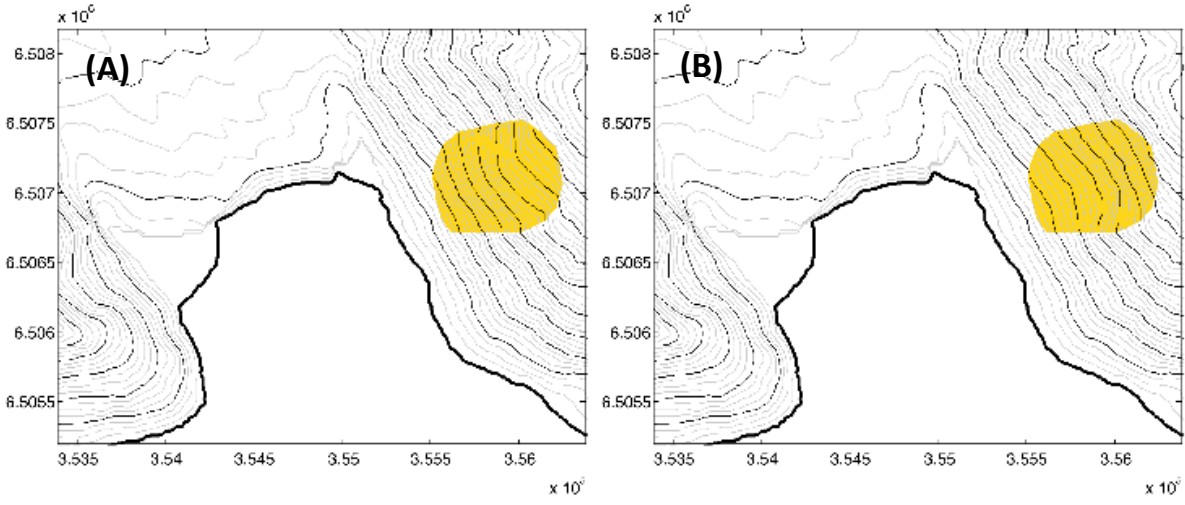

**Figure 4.** Details about the reconstructed slide. The isolines into the yellow area have been modified to reconstruct the location of the slide previous to the event. (A) Reconstructed topography of the east spur of Gilbert Inlet (before); (B) Topography of the east spur of Gilbert Inlet after the tsunami.

## 5.1 Landslide setup

In order to reproduce the main features of the slide impact, H. Fritz and collaborators designed a pneumatic landslide generator. They intend to model the transition from rigid to granular slide motion. Thus, at the beginning, the granular material is impulsed until the landslide achieves 110 m/s, that it is the approximated impact velocity between the slide and the water surface estimated by Fritz et al., assuming free fall equations for the centroid of the slide. From this instant, the slide is supposed to behave as a granular medium.

In this work, we have followed the same idea: assuming that 110 m/s is a good approximation for the impact velocity of the slide, an initial velocity for the granular layer has been estimated so that the computed impact velocity is approximately 110 m/s. This is a critical point for the performance of the simulation, in order to reproduce the dynamic of the impact, the generation of the tsunami, the propagation and the runup on different areas of the domain. Thus, for example, if model simulation is initialized from a solid slide at rest, this results in an impact velocity of the slide with the water of only, approximately, 67 m/s.

From a detailed analysis of the bathymetry surveys available for this study, an unexpected shifted location of the slide deposit on the floor of the Gilbert Inlet (see Figure 2.c), with a larger deposit concentration to the south part of the inlet was observed. The observation of this fact made us to consider a slide initial velocity vector, $v_s$, slightly shifted to the south, with modulus closed to 80 m/s (see Figure 3). Thus, with this initial condition, the model reproduces both the runup on the spur southwest of Gilbert Inlet and a giant wave traveling into the bay with enough energy to accurately reproduce the effects of the wave along the Lituya Bay.





# 6   Model description

Coulomb-type models for granular driven flows have been intensively investigated in the last decade, following the pioneering work of Savage-Hutter Savage and Hutter (1989), who derived a shallow water type model including a Coulomb friction term to take into account the interaction of the avalanche with the bottom topography. This model has been extended and generalized in several works (Bouchut et al. (2003), Bouchut and Westdickenberg (2004), Pelanti, Bouchut and Mangeney (2008), Bouchut et al. (2008), Bouchut et al. (2016)). In this framework, EDANYA group has implemented a finite volume numerical model for the simulation of submarine landslides based on the two-layer Savage-Hutter model introduced in Fernández-Nieto et al. (2008) that takes into account the movement of the fluid inside which the avalanche develops. This model is useful for the generation and evolution of tsunamis triggered by both submarine and aerial landslides.

This section describes the system of partial differential equations modelling landslide generated tsunamis based on layered average models. The 2D Landslide-HySEA model, is a two-dimensional version of the model proposed in Fernández-Nieto et al. (2008) for 1D problems, where local coordinates are not considered.

## 6.1   Simplified Two-layer Savage-Hutter type model

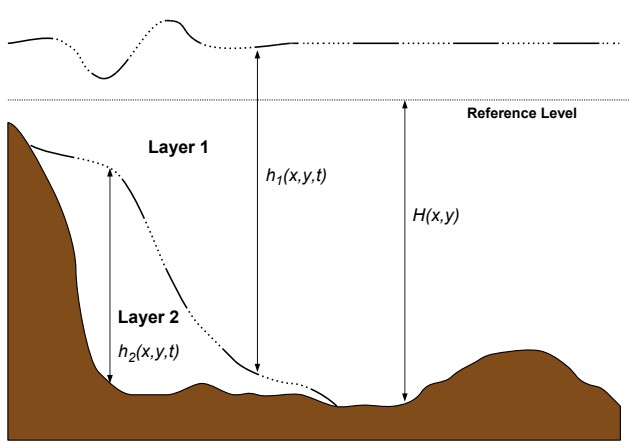

**Figure 5.** Sketch of the two-layer model. Relation among $h_1$, $h_2$ and $H$.

Let us consider a layered medium composed by a layer of inviscid fluid with constant homogeneous density $\rho_1$ (water), and a layer of granular material with density $\rho_s$ and porosity $\psi_0$. We assume that both layers are immiscible and the mean density of the granular material layer is given by $\rho_2 = (1 - \psi_0)\rho_s + \psi_0\rho_1$. The system of PDE describing the coupled two-layer system





writes as:

$$
\begin{cases}
\dfrac{\partial h_1}{\partial t} + \dfrac{\partial q_{1,x}}{\partial x} + \dfrac{\partial q_{1,y}}{\partial y} = 0 \\[3mm]
\dfrac{\partial q_{1,x}}{\partial t} + \dfrac{\partial}{\partial x}\left(\dfrac{q_{1,x}^2}{h_1} + \dfrac{g}{2}h_1^2\right) + \dfrac{\partial}{\partial y}\left(\dfrac{q_{1,x}q_{1,y}}{h_1}\right) = -gh_1\dfrac{\partial h_2}{\partial x} + gh_1\dfrac{\partial H}{\partial x} + S_{f_1}(W) \\[3mm]
\dfrac{\partial q_{1,y}}{\partial t} + \dfrac{\partial}{\partial x}\left(\dfrac{q_{1,x}q_{1,y}}{h_1}\right) + \dfrac{\partial}{\partial y}\left(\dfrac{q_{1,y}^2}{h_1} + \dfrac{g}{2}h_1^2\right) = -gh_1\dfrac{\partial h_2}{\partial y} + gh_1\dfrac{\partial H}{\partial y} + S_{f_2}(W) \\[3mm]
\dfrac{\partial h_2}{\partial t} + \dfrac{\partial q_{2,x}}{\partial x} + \dfrac{\partial q_{2,y}}{\partial y} = 0 \\[3mm]
\dfrac{\partial q_{2,x}}{\partial t} + \dfrac{\partial}{\partial x}\left(\dfrac{q_{2,x}^2}{h_2} + \dfrac{g}{2}h_2^2\right) + \dfrac{\partial}{\partial y}\left(\dfrac{q_{2,x}q_{2,y}}{h_2}\right) = -grh_2\dfrac{\partial h_1}{\partial x} + gh_2\dfrac{\partial H}{\partial x} + S_{f_3}(W) + \tau_x \\[3mm]
\dfrac{\partial q_{2,y}}{\partial t} + \dfrac{\partial}{\partial x}\left(\dfrac{q_{2,x}q_{2,y}}{h_2}\right) + \dfrac{\partial}{\partial y}\left(\dfrac{q_{2,y}^2}{h_2} + \dfrac{g}{2}h_2^2\right) = -grh_2\dfrac{\partial h_1}{\partial y} + gh_2\dfrac{\partial H}{\partial y} + S_{f_4}(W) + \tau_y
\end{cases}
\tag{1}
$$

In these equations, subscript 1 refers to fluid upper layer, and subscript 2 to the lower layer composed of the fluidized

material. $i$-th layer thickness at point $(x,y) \in D \subset \mathbb{R}^2$ at time $t$, where $D$ is the horizontal projection of domain occupied by the fluid,is denoted by $h_i(x,y,t)$. $H(x,y)$ indicates the depth of non erodible bottom measured from a fixed reference level at point $(x,y)$, and $\boldsymbol{q}_i(x,y,t) = (q_{i,x}(x,y,t), q_{i,y}(x,y,t))$ is the flow of the $i$-th layer at point (x, y) at time $t$, that are related to the mean velocity of each layer $(\boldsymbol{u}_i(x,y,t))$ by $\boldsymbol{q}_i(x,y,t) = h_i(x,y,t)\,\boldsymbol{u}_i(x,y,t)$, $i=1,2$. The value $r = \rho_1/\rho_2$ denotes the ratio between the constant densities of the two layers $(\rho_1 < \rho_2)$. Note that $H(x,y)$ does not depend on $t$, that is, the non-erodible

bottom topography does not change through the simulation although bottom may change due to second layer movement. Figure 5 graphically shows the relationship between $h_1$, $h_2$ and $H$. Usually, $h_1 + h_2 = H$ at rest or they represent the mean sea level.

The terms $S_{f_i}(W)$, $i = 1,\ldots,4$, model the different effects of dynamical friction, while $\boldsymbol{\tau} = (\tau_x, \tau_y)$ is the Coulomb friction law. $S_{f_i}(W)$, $i = 1,\ldots,4$, are given by:

$$
\begin{cases}
S_{f_1}(W) = S_{c_x}(W) + S_{a_x}(W); & \quad S_{f_2}(W) = S_{c_y}(W) + S_{a_y}(W); \\[3mm]
S_{f_3}(W) = -r\,S_{c_x}(W) + S_{b_x}(W); & \quad S_{f_4}(W) = -r\,S_{c_y}(W) + S_{b_y}(W).
\end{cases}
$$

The term $S_c(W) = \big(S_{c_x}(W), S_{c_y}(W)\big)$ parameterizes the friction between the two layers, and is defined as:

$$
\begin{cases}
S_{c_x}(W) = m_f \dfrac{h_1 h_2}{h_2 + r h_1}(u_{2,x} - u_{1,x})\,\|u_2 - u_1\| \\[4mm]
S_{c_y}(W) = m_f \dfrac{h_1 h_2}{h_2 + r h_1}(u_{2,y} - u_{1,y})\,\|u_2 - u_1\|
\end{cases}
$$





where $m_f$ is a positive constant, and $S_a(W) = \big(S_{a_x}(W),\, S_{a_y}(W)\big)$ parameterizes the friction between the fluid and the non-erodible bottom, and is given by a Manning law

$$
\begin{cases}
S_{a_x}(W) = -gh_1 \dfrac{n_1^2}{h_1^{4/3}}\, u_{1,x}\, \|u_1\| \\[2mm]
S_{a_y}(W) = -gh_1 \dfrac{n_1^2}{h_1^{4/3}}\, u_{1,y}\, \|u_1\|
\end{cases}
$$

where $n_1 > 0$ is the Manning coefficient.

$S_b(W) = \big(S_{b_x}(W),\, S_{b_y}(W)\big)$ parameterizes the friction between the granular and the non-erodible bottom, and as in the previous case, is given by a Manning law:

$$
\begin{cases}
S_{b_x}(W) = -gh_2 \dfrac{n_2^2}{h_2^{4/3}}\, u_{2,x}\, \|u_2\| \\[2mm]
S_{b_y}(W) = -gh_2 \dfrac{n_2^2}{h_2^{4/3}}\, u_{2,y}\, \|u_2\|
\end{cases}
$$

where $n_2 > 0$ is the corresponding Manning coefficient.

    Note that $S_a(W)$ is only defined where $h_2(x,y,t) = 0$. In this case, $m_f = 0$ and $n_2 = 0$. Similarly, if $h_1(x,y,t) = 0$ then

$m_f = 0$ and $n_1 = 0$.

    Finally, $\boldsymbol{\tau} = (\tau_x, \tau_y)$ is defined as follows:

$$
\text{If } \|\boldsymbol{\tau}\| \geq \sigma^c \Rightarrow
\begin{cases}
\tau_x = -g(1-r)h_2 \dfrac{q_{2,x}}{\|q_2\|}\, \tan(\alpha) \\[2mm]
\tau_y = -g(1-r)h_2 \dfrac{q_{2,y}}{\|q_2\|}\, \tan(\alpha)
\end{cases}
$$

$$
\text{If } \|\boldsymbol{\tau}\| < \sigma^c \Rightarrow q_{2,x} = 0, \quad q_{2,y} = 0
$$

where $\sigma^c = g(1-r)h_2 \tan(\alpha)$, where $\alpha$ is the Coulomb friction angle.

    System (1) can be written as a system of conservation laws with source terms and nonconservative products Fernández-Nieto

et al. (2008). In the next section, the finite volume scheme used to discretize system (1) is described. As friction terms are semi-implicitly discretized, we first consider that $S_F(W) = 0$. Then, the way those terms are discretized is briefly described.

## 7   Numerical scheme

To discretize system (1), the domain $D$ is divided into $L$ cells or finite volumes $V_i \subset \mathbb{R}^2$, $i = 1, \ldots, L$, which are assumed to be closed polygons. We assume here that the cells are rectangles with edges parallels to Cartesian axes. Given a finite volume $V_i$,

$N_i \subset \mathbb{R}^2$ is the center of $V_i$, $\aleph_i$ is the set of indices $j$ such that $V_j$ is a neighbor of $V_i$, $\Gamma_{ij}$ is the common edge of two neighboring volumes $V_i$ and $V_j$, and $|\Gamma_{ij}|$ is its length; $\boldsymbol{\eta}_{ij} = (\eta_{ij,x}, \eta_{ij,y})$ is the unit normal vector to the edge $\Gamma_{ij}$ and pointing towards $V_j$.

    We denote by $W_i^n$ an approximation of the solution average over the cell $V_i$ at time $t^n$:

$$
W_i^n \cong \frac{1}{|V_i|} \int_{V_i} W(x,y,t^n)\, dx\, dy
$$





where $|V_i|$ is the area of cell $V_i$ and $t^n = t^{n-1} + \Delta t$, where $\Delta t$ is the time step.

Let us suppose that $W_i^n$ is known. Thus, to advance in time, a family of one-dimensional Riemann problems projected in the normal direction to each edge of the mesh $\Gamma_{ij}$ is considered. Those Riemann problems are approximated by means of IFCP numerical scheme (see Fernández-Nieto et al. (2011)). Finally, $W_i^{n+1}$ is computed by averaging these approximate solutions. The resulting numerical scheme writes as follows:

$$W_i^{n+1} = W_i^n - \frac{\Delta t}{|V_i|} \sum_{j \in \aleph_i} |\Gamma_{ij}| \mathcal{F}_{ij}^-(W_i^n, W_j^n, H_i, H_j) \tag{2}$$

To check the precise definition for the numerical fluxes, $\mathcal{F}_{ij}^-(W_i^n, W_j^n, H_i, H_j)$, see Sánchez-Linares et al. (2015) or de la Asunción et al. (2016).

## 7.1 Wet/dry fronts

The numerical scheme described above, when applied to wet-dry situations, may produce incorrect results: The gradient of the bottom topography generates spurious pressure forces and the fluid can artificially climb up slopes. In Castro et al. (2005), to avoid this problem, a modification of the numerical scheme is proposed. Here the same strategy is to correct the proposed numerical scheme to suitably deal with wet-dry fronts. With this strategy, spurious waves reflection in the coast are avoided and a more realistic simulation of the flooded areas is obtained. Moreover, transitions between sub and supercritical flows, that appears continuously in simulations as the one presented here, which further complicate matters, are also suitably treated.

The implementation of the wet-dry front treatment in the numerical scheme results in not having to impose boundary conditions at the coasts. Coastline becomes a moving boundary, computed by the numerical scheme. Depending on the impact wave characteristics or the water back-flow movement, the computational cells are filled with water or they run dry, respectively. Consequently, no specific stabilization model technique is either required.

## 7.2 Friction terms discretization

In this section the numerical scheme when the friction terms $S_F(W)$ are discretized is presented. First, the terms $S_{f_1}(W)$, $S_{f_2}(W)$, $S_{f_3}(W)$ y $S_{f_4}(W)$ are discretized semi-implicitly; next the Coulomb friction term $\boldsymbol{\tau}$ will be discretized following Fernández-Nieto et al. (2008). The resulting numerical scheme is a three-step method, where in the first step, the IFCP scheme is used and then in the other two steps, the dynamical and static friction terms will be discretized. The three-step method will be denoted as follows:

$$W_i^n \to W_i^{n+1/3} \to W_i^{n+2/3} \to W_i^{n+1}$$

The resulting scheme is exactly well-balanced Sánchez-Linares et al. (2015) for the stationary water at rest solution ($\boldsymbol{q}_1 = \boldsymbol{q}_2 = \boldsymbol{0}$ and $\mu_1$ and $\mu_2$ constant). Moreover, the scheme solves accurately the stationary solutions corresponding to $\boldsymbol{q}_1 = \boldsymbol{q}_2 = \boldsymbol{0}$, $\mu_1$ constant and $\partial_x \mu_2 < \tan(\alpha)$ and $\partial_y \mu_2 < \tan(\alpha)$, that is a stationary water at rest solution for which the Coulomb friction term balances the pressure term in the granular material.





## 7.3 Numerical resolution and GPU implemetation

Landslide initial conditions have been described in Section 5.1. Initially water is at rest, and according to Miller (1960), an initial level of $1.52\,\text{m}$ have been set. The computational grid considered for the numerical simulation is a $4\,\text{m} \times 7.5\,\text{m}$ rectangular mesh composed of $3,650 \times 1,271$, i.e. 4,639,150, cells. The numerical scheme described in this section has been implemented in CUDA programming language in order to be able to run the model in GPUs. A high efficient GPU based implementation of the numerical scheme allows us to compute high accuracy simulations in very reasonable computational time.

The numerical simulation presented in this work covers a wall clock time period of 10 minutes. 14,516 time iterations were necessary to evolve from initial conditions to final state,10 minutes later. This required a computational time of 1,528.83 s (approx 25.5 min), which means 44 millions of computational cells processed per second in a nVidia GTX480 graphic card.

## 8 Parameter sensitivity analysis

To overcome the uncertainty inherent to the choice of model parameters and in order to produce a numerical simulation as close as possible to the real event, a sensitivity analysis has been performed. To do so, the three key parameters: (1) Coulomb friction angle, $\alpha$, (2) the ratio of densities between the water and the mean density of the slide, $r$, and, (3) the friction between layers $m_f$, have been retained as varying parameters for this sensitivity analysis. The values for these three parameters have been moved over the following ranges of reasonable values:

$$\alpha \in [10°, 16°] \qquad r \in [0.3, 0.5] \quad m_f \in [0.001, 0.1].$$

Four criteria were selected in order to get the optimal parameters:

**C1** the runup on the spur southwest of Gilbert Inlet had to be the closest to the optimal 524 m,

**C2** the wave moving southwards to the main stem of Lituya bay had to cause a peak close to 208 m in the vicinity of Mudslide Creek,

**C3** the simulated wave had to break through the Cenotaph Island, opening a narrow channel through the trees Miller (1960),

**C4** the trimline maximum distance of 1,100 m from high-tide shoreline at Fish Lake had to be reached.

Hundreds of simulations were performed in order to find the optimal values for the parameters that best verified the four conditions mentioned above. Finally, the optimal parameters found were:

$$\alpha = 13°, \qquad r = 0.44, \qquad m_f = 0.08.$$

Setting the three parameters to the values given above, simulation satisfied the previous four criteria with very good accuracy, more precisely:

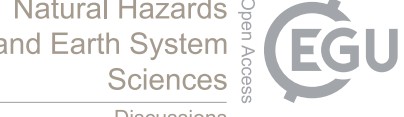



**Figure 6.** Images showing the four criteria used to select the optimal numerical solution and comparison of simulated optimal inundated area with the observed trimline. (A) Gilbert and Crillon Inlets; (B) Inner part of the Lituya Bay from Cenotaph Island; (C) Fish Lake and Cenotaph Island area; and (D) Entrance of the bay up to Cenotaph Island.

**SC1** the runup in Gilbert Inlet reached 523.85 m (Fig. 6.a);

**SC2** the runup peak in the vicinity of Mudslide Creek reached a height close to 200 m (Fig. 6.b);

**SC3** the wave produced a narrow channel crossing through Cenotaph Island (Fig. 6.c), and, finally,

**SC4** the runup reached more than 1,100 m distance from high-tide shoreline in Fish Lake area (Fig. 6.c).

5    Figure 6 graphically shows the four selected criteria enumerated above. Next section describes in some detail the numerical experiment performed with the optimal set of parameters.





## 9  Model Results

Sensitivity analysis provided the optimal set for the three key parameters considered for this study. In this section, model results corresponding to the simulation performed with these optimal parameters is presented: first the main characteristics of the giant wave generated in Gibert Inlet, second optimal description of wave evolution through the main stem of the Lituya Bay, and third, inundation details resulting from a comparison of numerical simulation runup with the real trimline observed in several areas of interest. Additional material in the form of a numerical simulation movie is also provided.

### 9.1  Giant wave generation and evolution in Gilbert Inlet. t: 0 s - 39 s

Following landslide trigger, the generated wave reaches its maximum amplitude, $272.4\,\mathrm{m}$, at $t = 8\,\mathrm{s}$ (Figure 7.b). The wave spreads outwards in a southward direction with decreasing amplitude (Figures 7.c, 7.d). $22\,\mathrm{s}$ after triggering, the wave hits the bottom of southwest spur of Gilbert Inlet. At $t = 39\,\mathrm{s}$ the maximum $523.9\,\mathrm{m}$ runup is reached (Figure7.f).

### 9.2  Wave evolution

### t: 30 s - 2 min

While the maximum runup on the east side of Gilbert head is reached, the southern propagating part of the initial wave, with a height of more than $100\,\mathrm{m}$, moves in a south-west direction, hitting the south shoreline of the Bay after $35\,\mathrm{s}$ (Figure 8.a, 8.b). The impact causes maximum runup close to $180\,\mathrm{m}$ to occur in the vicinity of Mudslide Creek at $t = 70\,\mathrm{s}$ (Figure 8.c, 8.d). In the meanwhile, part of the water reaching the maximum runup area over Gilbert head retreats and part flows over Gilbert head, inundating the observed affected area to the south.

### t: 2 min - 3 min

While the initial wave propagates through the main axis of Lituya Bay to Cenotaph Island, a larger second wave appears as reflection of the first one from the south shoreline (Figure 8.c, 8.d). Both waves swept each of the shorelines in their path. Along the north shoreline, wave runup reaches between $50\text{-}80\,\mathrm{m}$ in height (8.e) while in along the south shoreline, the runup reaches heights between $60\text{-}150\,\mathrm{m}$.

The first wave reaches Cenotaph Island after $2\,\mathrm{min}$ and $5\,\mathrm{s}$ with a mean amplitude close to $20\,\mathrm{m}$ (Figure 8.e,f) flooding over more than $650\,\mathrm{m}$ from the most eastern prominence of the island and about $700\,\mathrm{m}$ from the little cape, slightly southwards. About 25 seconds later, a second wave of approximately $32\,\mathrm{m}$ height hits the east coast (Figure 8.g, 8.h).

### t: 3 min - 5 min

After hitting Cenotaph Island, the wave splits into two parts; one advances in the shallow channel north of the island and the second travels through the deeper channel south of the island (Figure 9.a, 9.b). Waves higher than $25\,\mathrm{m}$ hit the north shoreline area in front of Cenotaph Island causing large extent runups (above $1\,\mathrm{km}$ inland from the coastline) in the east area near Fish Lake (Figure 9.c, 9.d). Along the south shoreline in front of Cenotaph Island, larger waves with $40\text{-}50\,\mathrm{m}$ of amplitude hit the coast and penetrate about $1\,\mathrm{km}$ inland on the flat areas located east of the Paps (Figure 9.e, 9.f).



**Figure 7.** Giant wave generation and initial evolution in Gilbert Inlet. (A) $t = 0$ s.; (B) $t = 8$ s. The giant wave reaches 272.4 m amplitude; (C) $t = 10$ s. Maximum wave amplitude reaches 251.1 m.; (D) $t = 20$ s. Maximum wave amplitude reaches 161.5 m; (E) $t = 30$ s. Giant wave hits the spur southwest of Gilbert Inlet; (F) $t = 39$ s. Maximum runup: 523.9 m.

**t: 5 min - 8 min**

Large inundated areas were formed both around Fish Lake, in the north shoreline, as in the flat areas surrounding the Paps, while the main wave reaches the narrow area near La Chausse Spit. During this time, the wave amplitude is larger than 15 m, indeed over 20 m on the north shoreline. At 5 min 50 s the wave reaches La Chausse Spit, passing over before reaching the sea, and then partially reflecting a wave back into the bay (Figure 9.g, 9.h).





**Figure 8.** Wave evolution from $t = 40$ s. until $t = 2$ min. $t = 30$ s. In the right hand panels the track of the inundated areas is kept. Left panels for South view, right panels for plan view.

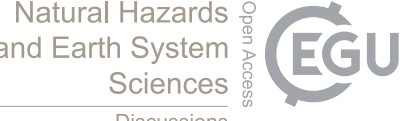

**Figure 9.** Wave evolution from $t = 3$ min 30 s until $t = 7$ min. In the right hand panels the track of the inundated areas is kept. Left panels for South view, right panels for plan view.





### 9.3 Inundation assessment

In this section, the computed inundated area limit is compared with the real trimline drawn by Miller. Figure 10 depicts the total extent of the inundated area and the maximum water height reached. The trimline determined by Miller (Figure 1) is superimposed in pink. In order to closely assess the accuracy of the predicted numerical inundated area with respect to the

observed trimline, we have performed this comparison focusing successively on the different areas of interest.

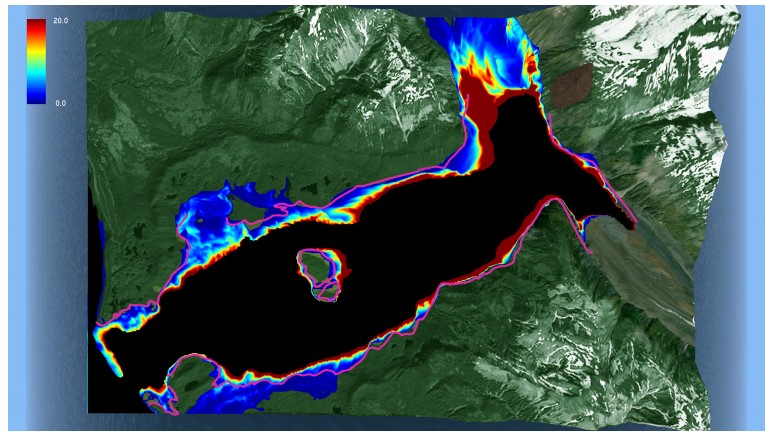

**Figure 10.** Simulated inundated areas and maximum height all around the Lituya Bay. Trimline in pink.

### 9.3.1 Gilbert Inlet

As it has been shown, on Gilbert Head, the maximum runup (523.85 m) is reached on the east slope. Furthermore, the runup is extended oblique-to-slope on the western face of Gilbert head. Runup extent and trimline coincide quite remarkably in this sector (Fig. 6.a). There is large extent inundated area over the Lituya Glacier, from its shoreline up to more than 2 km over

the glacier. Finally, a good correlation between trimline and inundation on the east slope of Gilbert Inlet where the slide was initially located is observed.

### 9.3.2 North Shoreline

Figure 10, shows the good agreement between the model simulated inundated area and the real trimline around the east part of the north shore due to the higher slopes. Good agreement between trimline and simulated runup in the north shoreline of La

Chausse Spit is also found.

### 9.3.3 Fish Lake area

The agreement between model and observation around the flat areas surrounding Fish Lake is good (Fig. 6.c). Here, the inundation extent includes vicinity areas of Fish Lake under 40 m height. In order to achieve a better agreement between





simulated inundation limit and observed trimline, it would be necessary to consider a map of drag friction to account for different friction coefficients depending on the type of vegetation or soil Kaiser et al. (2011). In flat areas, good agreement between model results and observed data requires that the presence of vegetation or any other type of obstacle not collected in the topography be taken into account or parameterized in some way.

### 9.3.4 South Shoreline

The computed runup underestimates the trimline over steeper slopes in the eastern third of the south shores (Fig 10). Moreover, the numerical model provides mean runup heights around $120\,\text{m}$ while trimline heights move from $140$ to $200\,\text{m}$ around this sector. The reason for this mismatch is probably due to numerical resolution. In order to capture the steep slopes in this area, a higher numerical resolution than provided by available data would be required. There is good agreement along The Paps shores as noted on the south shoreline in front of La Chausse Spit (Fig. 6.d).

### 9.3.5 East area of The Paps

Around the central third of the south shoreline, the agreement between model and observed trimline is less coherent (Fig. 6.d). On the flat east area near The Paps, large inundation occurs, flooding the sector area with a height of $30 - 40\,\text{m}$. As already mentioned, it would be necessary to consider a map of drag frictions in order to achieve more precise results over this area.

### 9.3.6 Cenotaph Island

One of the items to be checked in the sensitivity analysis presented in Section 8 was related to flooding on Cenotaph Island. Therefore good agreement is expected in this particular location. In fact the trimline and the computed inundation limit closely track on the island as shown in Figure 6.c.

### 9.3.7 La Chausse Spit

As has been previously described, La Chausse Spit was completely covered by water from minute 6 for more than 90 seconds (Figs. 9.g, 9.h). The computed inundation around La Chausse Spit is in good agreement with the observed trimline (Fig. 6.d).

## 10 Discussion

### 10.1 Potential sources of error

The Landslide-HySEA model was tested against analytic solutions and laboratory measurements of Fritz et al. (2009) (Sánchez-Linares (2011), and work in progress). In addition, Landslide-HySEA recently participated in the "NTHMP/MMS Landslide Model Benchmarking Workshop" hosted by Texas A&M University at Galveston on 9-11 January 2017 during which the model satisfactorily reproduced expected results. Landslide-HySEA is a finite volume coupled landslide-fluid model that acts as a shallow water model when the slide layer is immobile or when there is not a sediment layer in the column of water. A suitable




and simple treatment of the wet-dry fronts avoids spurious wave reflection on the coast and produces a realistic simulation of flooded areas. Transitions between sub- and super-critical flows, that continuously appear in simulations including those presented here, and which further complicate matters, are also suitably treated.

Nonetheless, as it was previously mentioned, potential sources of model errors are the quality of model initialization pa-
rameters, the initial landslide conditions, or the Digital Elevation Model (DEM) due to limitations associated with bathymetric data. Moreover, in real landslides, the material is neither homogeneous or granular, as assumed here in the present study. Nevertheless, this type of model can be used in practice to provide general information about the generated tsunami and the flooded areas as demonstrated in the presented results.

### 10.1.1   Limitations of the DEM and Digital Bathymetry

A high quality DEM is necessary to properly model tsunami wave dynamics and inundation onshore, especially in areas with complicated bathymetries. In this study there was an additional need for good information on topo-bathymetric data just before the event in order to produce realistic, pre-event geometry.

Though we have combined the DEM based on the best available data in the region (described in Section 4), neither pre-tsunami bathymetry data of the bay nor the definition of Lituya Glacier front just before the 1958 tsunami were available with
fine enough resolution or quality. Of event greater significance, estimations of the volume and position of the slide that caused the tsunami were all that were available. Thus, as was described in Sections 3 and 5, a proposed reconstruction of the original Lituya Glacier shoreline provided by Miller (1960) and data from the 1926 & 1959 U.S. Coast and Geodetic Surveys were used in this study.

### 10.2   Model results

Due to the choice of optimal parameters in the sense described in Section 8, the simulation presented achieves the main objectives proposed. Section 9 presents the first stage of the tsunami dynamics in Gilbert Inlet: Giant wave generation and the inundation induced over the east slope of Gilbert Inlet. Later, the wave propagation in south-west direction along the Lituya Bay is described until the wave crosses La Chausse Spit. Model results are in good agreement with those described in Miller (1960).

In a second study stage, an inundation assessment is performed. A detailed description of the runup areas along the shores of the bay is presented. In general, computed inundation areas are in very good agreement with Miller observations. Nevertheless, the model provides larger inundation areas than the $10.35\,\mathrm{km}^2$ between the trimlines and the high-tide shorelines estimated by Miller. It is noted, however, that Miller made an estimate of the total area inundated by the wave of at least $13\,\mathrm{km}^2$, an estimate that is closer to the model results.



## 11 Conclusions and Future work

It has been demonstrated that the landslide triggering mechanism proposed by Fritz et al. (2009) is crucial in order to reproduce not only the wave dynamics inside Gilbert Inlet, but also all tsunami dynamics produced along the bay, including inundation effects, wave heights, and several details observed in Miller (1960). The simulated wave heights and runup (as assessed by the 5 trimline location) are in good agreement with the majority of observations and conclusions described by Miller (1960).

It has been shown that the numerical model used can simulate subaerial scenarios similar to the Lituya Bay's case provided that some information is available to calibrate the model. The main question that remains to be answered is then obvious: what happens when information to calibrate the model is not available? In that case, which approach is followed? In another words, how an actual risk assessment study would be performed without post event information. In that case two approaches can be 10 followed. One first option is a deterministic approach, in which, depending on the characteristics of the slide, some coefficients are selected as default. In the case considered in this work, consisting in a slide moving, essentially as a solid block, the "blind" proposed parameters would be (in parentheses the optimal values for comparison):

$$\alpha = 16°(13°), \qquad r = 0.3(0.44), \qquad m_f = 0.08(0.08).$$

The numerical results for this case, that must be compared with the optimal case selected in Fig. 6, are presented in Figure 11. As can be observed, in this case, blind numerical results produce higher runup and larger inundated areas than the optimal 15 calibrated simulation, therefore, from the point of view of risk assessment, considering the standard parameters should not underestimate the risk. Even, in some areas the agreement is better than for the optimal case, although the four required criteria at the same time are better achieve in the optimal case.

Concerning future work, as uncertainty in the data (initial condition, model parameters, etc.) is of paramount importance in real applications, a promising line of research is uncertainty quantification. Therefore, some information of the main prob- 20 abilistic moments should be provided. Uncertainty quantification is currently a very active area of research, with one of the most efficient techniques utilized being multilevel Monte Carlo methods. To run such a method, a family of embedded meshes is first considered. Then, a large enough number of samples of the stochastic terms are chosen and, for each sample, a deterministic simulation is run. Finally, the probabilistic moments are then computed by a weighted average of the deterministic computations Sánchez-Linares et al. (2016).

25 Another improvement of the model envisioned will be carried out by considering shallow Bingham dense avalanche models, like those introduced in Fernández-Nieto et al. (2010), that will be coupled with the hydrodynamic model. In the other hand, current efforts are focused on the implementation of a model including dispersive effects to allow for comparison of model performance with and without dispersion Macías et al. (2017a). This effort too will serve to assess the role played by dispersive terms in these landslide generated types of events. In any case, the present work shows that a Savage-Hutter model coupled 30 with shallow water equations is sufficient to suitably reproduce the main features of an extreme event such as the one occurring in Lituya Bay in 1958.



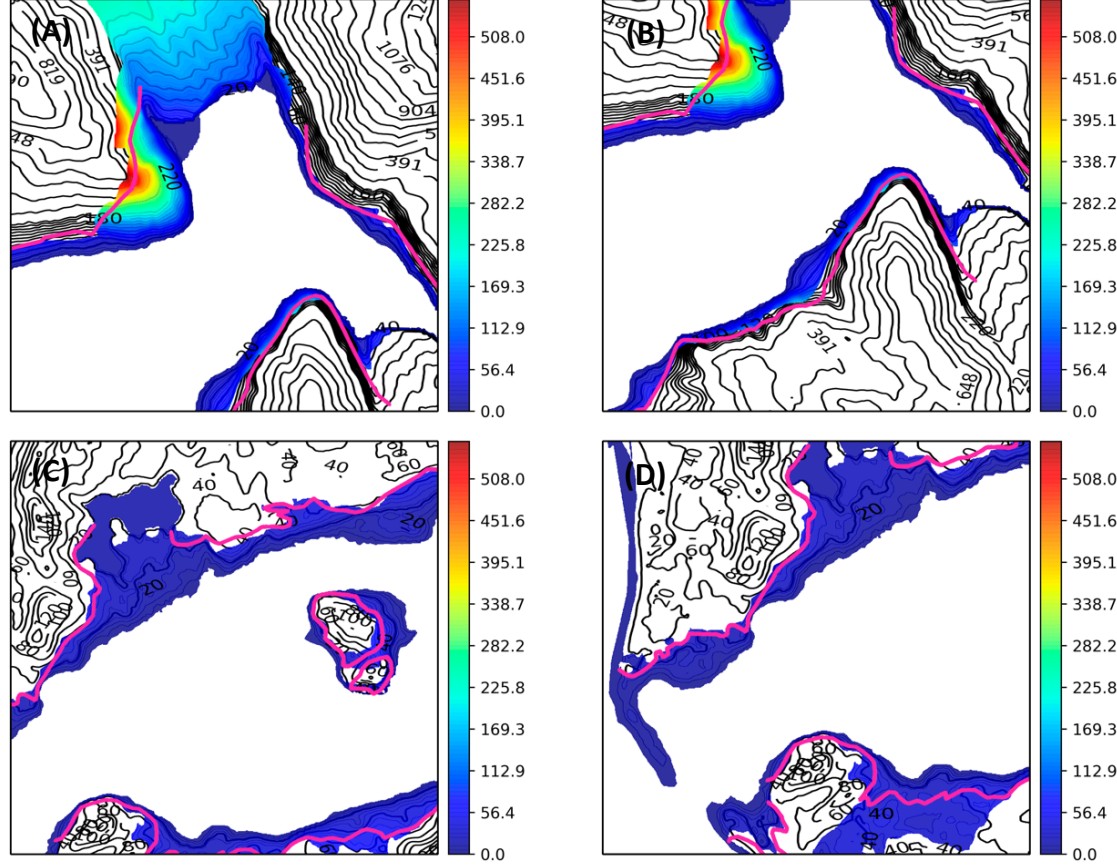

**Figure 11.** Same as in Fig. 6 but for the blind simulation with standard not optimally adjusted parameters. Comparison with the observed trimline. (A) Gilbert and Crillon Inlets; (B) Inner part of the Lituya Bay from Cenotaph Island; (C) Fish Lake and Cenotaph Island area; and (D) Entrance of the bay up to Cenotaph Island.

*Acknowledgements.* All numerical experiments required for the development of this research have been performed at the Unit of Numerical Methods of the University of Malaga. This work was partially funded by the NOAA Center for Tsunami Research (NCTR), Pacific Marine Environmental Laboratory (U.S.A.) Contract. No. WE133R12SE0035, by the Junta de Andalucía research project TESELA (P11-RNM7069), by the Spanish Government Research project SIMURISK (MTM2015-70490-C02-01-R) and Universidad de Málaga, Campus de Excelencia Andalucía TECH. All data required to perform the numerical simulations presented in this study are described in the text or are publicly available. D. Arcas was supported by the National Oceanic and Atmospheric Administration with this report being PMEL contribution No. 4665.



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
