# Peer review of "The Lituya Bay landslide-generated mega-tsunami. Numerical simulation and sensitivity analysis"

_Natural Hazards and Earth System Sciences, 2018_

## Referee Comment (RC1) · Anonymous Referee #1 · 2 Oct 2018

The paper deserves to be published for 2 reasons: - the paper is a good review of the 1958 event - the simulation attempts at modelling this event and the authors provide landslide parameters for reproducing the inundation

My main criticisms:

- I do not understand how to justify the initial velocity of the landslide (80m/s or 280km/h !!). Line 14 (5.1/Landslide setup) What are the references to justify such a velocity? In my opinion, the landslide model should reproduce the the landslide velocity from rest to the end of landslide. Otherwise, why do the authors model the landslide ? The justification of the authors (line 15) : "with this initial condition, the model reproduces

both the runup and ... a giant wave" is not valid in my opinion.

-There is no references in the literature to justify the 3 parameters for reproducing the water waves (the friction angle (alpha) , the ratio of densities (r) and the friction between layers (mf)). The authors have to cite references in particular on the friction angle.

- Friction of water : do the authors model this friction? I do not find the value of the friction coefficient. The authors have also to justify the choice of this coefficient or to discuss it.

- Sensitivity tests: "hundreds" of simulations have been performed according to the authors. At least, 3 graphs have to be added to show the influence of the friction angle, r and mf. Without these graphs (or tables), results are not seriously discussed.

- the authors have to justify the use of a nonlinear shallow water model. In this particular case, is it valid to neglect the vertical acceleration of water? in the generation area (in the Gilbert Inlet) and in the propagation area (in the Lituya Bay)?

- the introduction is lengthy. Some phrases are redundant and have to suppressed

Minor corrections: - the volume is not 31 km3 but 31 Mm3! (line 31/ Background) - check how to cite papers in the manuscript (for instance, p9 line5, p11 line 14 , ...) - a few spelling mistakes (for instance, p22 line17 : achieved/ p22 line 8 : other/p13 line2 : has/p12 line13: reflections/p5 line 28 order of/)

---

## Referee Comment (RC2) · Anonymous Referee #2 · 19 Nov 2018

The manuscript entitled "The Lituya Bay landslide-generated mega-tsunami. Numerical simulation and sensitivity analysis" submitted by Gonzalez-Vida and co-authors concerns an attempt to accurately reproduce numerically the generation, propagation and inundation of the famous historical 1958 Lituya Bay landslide-triggered tsunami. I generally find the manuscript very well written, well-structured, solidly argued, and carefully supported by appropriate figures. I also find the methodologies generally well-explained and with enough details on the numerical approach employed by the authors, as well as reasonable explanation concerning the assumptions and simplifications in that approach. The modelling results presented in this paper are able to successfully reproduce the run-up observed at Lituya bay, and the parameters used

are generally in agreement with observations and assumptions by Miller (1960) and Fritz et al. (2009). More important, the authors do a very good job at discussing and analysing the range of parameters that generate numerical solutions that are able to reproduce the observational constraints – and this is a plus in this paper. I am therefore of the opinion that the constitutes a timely and solid contribution to the field, offering a modern and sober analysis on the application of numerical solutions to reproduce this kind of events. In my view, the manuscript is worthy of publication following some very minor reviews. My main criticism concerns:

1) Referencing/citation style needs to be revised, given it is confusing and not in the right format in places, e.g. without parenthesis when they should have. Here is an example of this:

"This is particularly true for the leading wave Løvholt et al. (2015) that, on the other hand, is mainly responsible for coastal impact."

instead of

"This is particularly true for the leading wave (Løvholt et al., 2015) that, on the other hand, is mainly responsible for coastal impact."

Or

"It is in the far field where dispersive effects are proven to be important for a realistic description of tsunami impact Løvholt et al. (2008); Montagna et al. (2011)"

When it should be

"It is in the far field where dispersive effects are proven to be important for a realistic description of tsunami impact (Løvholt et al., 2008); Montagna et al., 2011)"

(If the authors used latex to prepare the manuscript perhaps they used the command \citet{author} instead of \citep{author}?)

2) an introduction that is perhaps overly long and a bit wordy – I guess the authors

could trim or synthesise this part of the text to make it easier for the potential reader.

---

## Author Response (AR2)

**ANSWER TO REVIEWER #1**

First of all, we would like to thank the very positive review from Referee #1.

Regarding his/her comments:

1) Concerning the question about the initial slide velocity, we tried to numerically reproduce the Fritz et al's laboratory experiment as we tried to point out at several places along the paper. We have included a sentence at the end of section 2 in order to make this more explicit, in particular, we wrote: *"Based on the experimental work of Fritz et al, in the present numerical study we will follow the same approach: an initial slide speed (analogous to the impulse of the pneumatic landslide generator in the lab experiment) will be imposed in order to get the 110 m/s slide impact velocity that Fritz et al measure in their experiment. The same way the laboratory experiment did reproduce the observed run-up, is the way the numerical experiment has been initialized."* Therefore, the references justifying this approach are the experimental works of Fritz et al.

2) **References for the 3-parameter models**. We think we do not understand precisely referee's comment. In the model used in the present work 5 parameters are required: 1) r, ratio of densities; 2) mf, friction coefficient used in the friction between the water and the slide; 3) n1, Manning coefficient for water/bottom friction parameterization; 4) n2, Manning coefficient for slide/bottom friction parameterization; and 5) alpha, the Coulomb friction (static) angle. We have chosen for the sensitivity study 3 of these parameters (r, mf, and alpha). Therefore, it should be more precise to refer as a "5 parameter model" for which we have retained the two Manning coefficients as constant values and the other 3 parameters have been varied for the sensitivity analysis. In any case, we have included several references to models using the same kind of friction parameterizations as the ones used here, as we think this is the point that the reviewer wanted to highlight.

   References included:
   - For the parameterization of the term Sc (water/granular slide interface) we used a particular case of Pitman and Le (2005) or Pelanti et al (2008) parameterization. We ha added these two references.
   - Dyakonova and Khoperskov (2018) for the Manning parameterization and Arcement and Schneider (1989) and Phillips and Tadayon (2006) for Manning coefficient values.
   - Savage and Hutter (1989) and Gray et al (1999) for Coulomb law

3) **Friction of water**. All possible frictions are modelled: (a) interface water-slide (Sc) with the mf coefficient, (b) water-bottom (Sa); and (c) slide-bottom (Sb). These three are the dynamical friction terms. For the friction with the bottom, (b) and (c), a Manning law is used, and as the reviewer points out, we did not give the explicit values set for the two Manning coefficients (now included). Besides, there is a static Coulomb friction term for the granular slide.

We have provided the value for $n_1=0.02$. It is a quite standard value used for this coefficient for water and sea bottom, usually varying between 0.015 and 0.03 in shallow water models. For river and channels tables can be found as in:
https://www.engineeringtoolbox.com/mannings-roughness-d_799.html
https://www.engineersedge.com/fluid_flow/manning-constant.htm

We have added the reference *Arcement and Schneider (1989)and Phillips and Tadayon (2006)* to justify the use of that value.

The value for $n_2$ was set to 0.05, as correspond to a larger friction between the slide and the non-erodible bottom.

We have added the sentence: *"The other two parameters required in model parameterizations, the Manning coefficients $n_1$ and $n_2$, were set constant with standard values of $n_1=0.02$ and $n_2=0.05$ \citep{Arcement_Schneider_1989, Phillips_Tadayon_2006}."* in Section 8.

4) **Sensitivity tests**. This was the most difficult item to give a suitable answer to the reviewer's comments, but I think the result deserved the effort. The reviewer is absolutely right when he/she points out the interest of a sensibility analysis, this provide the paper an added value and it is interesting *per se.* The reason why we did not perform and we did not include in the paper was that some of the criteria used
   - were not given by a number (C3) or
   - were difficult to measure and mostly to do so automatically (C4).
To overcome this problem and fulfil reviewer's request we have considered 4 regions (A, B, C, and D) and we have measured the maximum runup in these 4 regions. Doing so, criteria 1 and 2 are considered by runups in regions A and B respectively; in some sense criteria 3 is quantify by computing the runup in Cenotaph Island (region C), and criteria 4 is not considered and substituted by the maximum runup in a fourth region, D, closer to the exit of the Bay. Then we perform numerical experiments for a reduced number of the parameters considered for this study, the "macroscopic" set of parameters, composed by, for $r=$ 0.3, 0.4, 0.5 and 0.6; for $\alpha =$ 8°, 10°, 12°, 14°, 16°; and for $m_f =$ 0.001, 0.005, 0.01, 0.025, 0.05, 0.075, 0.1. Summing up a total of 180 numerical experiments. For all these experiments we have measured the maximum run-up in the coastal strip in each of the 4 regions considered and used these data to generate 3x4 graphs that we have gathered in three figures. I must acknowledge reviewer's comment as I think it has improved the quality of our work.
We have included the figures and the text required to explain all this.

5) Justification for using NLSW models. Part of the introduction is devoted to this aim, and confinement is one of the arguments, we dedicate some paragraphs to address this particular issue, and we also mention NTHMP agreement on the fact that NLSW models remain a suitable tool in enclosed basins as fjords. But another important argument used is that no previous more comprehensive model has been able to perform a numerical simulation of the complete 3D

Lituya Bay problem and we do prove that a NLSW model in what respect inundation area and run-up does a very good job.

6) The introduction has been slightly reduced in 12-14 lines.

All minor corrections have been included in the new version of the manuscript.

Finally, we would like to acknowledge this very positive and constructive review.

Jorge Macías on behalf of all coauthors

**ANSWER TO REVIEWER #2**

First of all, we would like to thank the very positive review from Referee #2.

Regarding his/her comments:

1) Referencing/citation style needs to be revised, given it is confusing and not in the right format in places, e.g. without parenthesis when they should have. Here is an example of this:
"This is particularly true for the leading wave Løvholt et al. (2015) that, on the other hand, is mainly responsible for coastal impact."
instead of
"This is particularly true for the leading wave (Løvholt et al., 2015) that, on the other hand, is mainly responsible for coastal impact."
Or
"It is in the far field where dispersive effects are proven to be important for a realistic description of tsunami impact Løvholt et al. (2008); Montagna et al. (2011)"
When it should be
"It is in the far field where dispersive effects are proven to be important for a realistic description of tsunami impact (Løvholt et al., 2008); Montagna et al., 2011)"
(If the authors used latex to prepare the manuscript perhaps they used the command \citet{author} instead of \citep{author}?)

The reviewer is absolutely right, I misused the LaTeX command for citation. This has been corrected all along the paper.

2) an introduction that is perhaps overly long and a bit wordy – I guess the authors could trim or synthesise this part of the text to make it easier for the potential reader.

This same comment has been made by the first reviewer and we have shortened the introduction.

In the revised version of our manuscript we have addressed the two items highlight by the reviewer:

1) The issue with the references due to a misuse of the \cite vs \citep LateX command
2) The length of the introductory section that have been reduced in 12 lines.

Jorge Macías on behalf of all coauthors

[revised manuscript text omitted]